# Tunable Thermal Camouflage Based on GST Plasmonic Metamaterial

**DOI:** 10.3390/nano11020260

**Published:** 2021-01-20

**Authors:** Qianlong Kang, Dekui Li, Kai Guo, Jun Gao, Zhongyi Guo

**Affiliations:** School of Computer and Information, Hefei University of Technology, Hefei 230009, China; 2018110992@mail.hfut.edu.cn (Q.K.); 2019170964@mail.hfut.edu.cn (D.L.); kai.guo@hfut.edu.cn (K.G.); gaojun@hfut.edu.cn (J.G.)

**Keywords:** thermal camouflage, tunable thermal emitter, plasmonic metamaterial, phase change material

## Abstract

Thermal radiation control has attracted increasing attention in a wide range of field, including infrared detection, radiative cooling, thermal management, and thermal camouflage. Previously reported thermal emitters for thermal camouflage presented disadvantages of lacking either tunability or thermal stability. In this paper, we propose a tunable thermal emitter consisting of metal-insulator-metal (MIM) plasmonic metamaterial based on phase-change material Ge_2_Sb_2_Te_5_ (GST) to realize tunable control of thermal radiation in wavelength ranges from 3 μm to 14 μm. Meanwhile, the proposed thermal emitter possesses near unity emissivity at the wavelength of 6.3 μm to increase radiation heat dissipation, maintaining the thermal stability of the system. The underlying mechanism relies on fundamental magnetic resonance and the interaction between the high-order magnetic resonance and anti-reflection resonance. When the environmental background is blackbody, the tunable emitter maintains signal reduction rates greater than 80% in middle-IR and longer-IR regions from 450 K to 800 K and from room temperature to 800 K, respectively. The dependences of thermal camouflage on crystallization fraction of GST, incident angles and polarization angles have been investigated in detail. In addition, the thermal emitter can continuously realize thermal camouflage for various background temperatures and environmental background in atmospheric window in the range of 3–5 μm.

## 1. Introduction

The ability of controlling thermal radiation is important in a broad range of applications, including radiative cooling [1,2,3], infrared detection [4,5], thermophotovoltaics [6,7,8], thermal camouflage [9,10,11,12,13,14,15], and thermal management [16,17]. In recent times, the metamaterial consisting of a series of periodic or regionally non-periodic arrayed unit cells has attracted much attention in the field of electromagnetic wave manipulation [18,19,20,21,22,23]. For example, the plasmonic nanostructures can be applied as an absorber that achieves near unity perfect absorption by electromagnetic resonance modes. Through flexibly designing geometric structures and patterns, the location, number, strength, and bandwidth of absorption peak can be desirably engineered from visible to terahertz spectrum [24,25,26]. According to Kirchhoff’s thermal radiation law, absorptivity and emissivity of an object are equal in the aspect of spectrum, angle and polarization. Thereby, the absorbers based on plasmonic nanostructures have special ability to control thermal radiation. Up to now, several plasmonic nanostructures including metal-insulator-metal (MIM) structures [27,28], all metallic structures [29] and gratings [30] have been used to control thermal radiation. Generally, thermal emitters based on plasmonic nanostructures are ultrathin because incident light can be confined in nanoscale volume of plasmonic nanostructures, leading to an enhanced electromagnetic field. However, they only present a static manipulation of thermal radiation without flexible tunability, restricting their application prospects. To realize dynamical controls of thermal radiation, several functional plasmonic devices based on phase-change material Ge_2_Sb_2_Te_5_ (GST) are proposed in the past [31]. GST is non-volatile and reconfigurable, and can be continuously modulated between amorphous states and crystalline states at different temperatures and stimulation [32,33,34].

Thermal camouflage is important in military tactics, which is usually utilized by warships and submarines to make them invisible for IR detectors and thermal cameras. With the fast development of IR detection technology, dynamical control of thermal radiation is a critical issue in thermal camouflage of military objects and critical targets as external circumstances changes. To date, several thermal camouflage devices have been practically presented and experimentally demonstrated in aspects of thermal camouflage performance. By electrically tuning charge density of graphene, the emissivity of a thermal emitter consisting of a multilayer-graphene electrode on a porous polyethylene (PE) membrane and a back gold-electrode can be dynamically tailored in a broadband wavelength range from 5 μm to 25 μm, achieving adaptive thermal camouflage [35]. Using a simple actuating mechanism, geometric microstructures of the system can be dynamically modulated to realize thermal camouflage at wavelength of 3–5 μm, which is work wavelength of IR detector [36]. By thermally controlling the phase-change material like GST, the thermal emitter based on GST-Au double-layered film can dynamically tune emissivity from 5 μm to 14 μm for near perfect thermal camouflage [37]. Overall, these thermal camouflage devices lack the ability to increase radiation heat dissipation at undetected band of 5–8 μm.

In this paper, a tunable thermal emitter, consisting of MIM metamaterial based on phase-change material GST, is theoretically investigated to realize tunable control of thermal radiation and thermal camouflage in both atmospheric windows. Meanwhile, the tunable thermal emitter has near unity emissivity at wavelength of 6.3 μm to improve radiation heat dissipation, maintaining the thermal stability of the system. When the environmental background is blackbody, the tunable emitter maintains signal reduction rates greater than 80% in middle-IR and long-IR regions from 450 K to 800 K and from room temperature to 800 K, respectively. Subsequently, the dependence of thermal camouflage on crystallization fraction of GST, incident and polarization angles have been also investigated. In addition, the thermal emitter can continuously realize thermal camouflage for varies background temperatures and environmental background in atmospheric window ranges from 3 μm to 5 μm. Our proposed thermal emitter will pave the way towards the tunable thermal radiation control, thermal management and thermal camouflage in both civilian and military applications.

## 2. Structure Design

Thermal camouflage is achieving low observability of vital targets by decreasing the IR radiation intensity of object itself. As shown in Figure 1a, IR detectors can receive the thermal radiation of an object itself and the solar reflection. Figure 1b depicts the schematic diagram of tunable ultrathin MIM plasmonic thermal emitters, and Figure 1c presents a corresponding cross-section view of a unit cell. The thicknesses of the bottom gold layer, the intermediate GST layer and the top gold nanodisks are set as *t*_3_ = 100 nm, *t*_2_ = 240 nm and *t*_1_ = 100 nm, respectively. The array periodicity is *p* = 3 μm, and the diameter of the nanodisks is *d* = 1.5 μm. The deposited GST is in an amorphous phase (called aGST). An annealing process above phase-change temperature on a hot plate is applied to obtain GST in crystalline phase (termed as cGST). It should be noted that the interlayer diffusion between GST and gold disks exists in the annealing process [38]. Herein, we only consider ideal condition without the interlayer diffusion between GST and gold disks in our simulations. The simulated spectral emissivity is obtained at normal incidence, as shown in Figure 1c, except for special illustrations. According to Kirchhoff’s law of thermal radiation, the spectral emissivity can be obtained by calculating the spectral absorptivity of the MIM thermal emitter. Therefore, the emissivity of the MIM thermal emitter is numerically investigated by using finite-difference time-domain method. The relative permittivity of gold and GST (3–14 μm) is obtained from Palik’s handbook [39] and Ref. [40], respectively.

The GST layer in the intermediate stage is composed of amorphous and crystalline molecules in different proportions. In terms of numerical simulation, the effective permittivity ε_eff_ (λ) of different intermediate GST phases can be estimated through effective-medium theories [41] based on the Lorentz–Lorenz relation [42,43,44]:(1)$$\frac{\varepsilon_{\mathrm{eff}}(\lambda )-1}{\varepsilon_{\mathrm{eff}}(\lambda )+2}=m\times \frac{\varepsilon_{c}(\lambda )-1}{\varepsilon_{c}(\lambda )+2}+(1-m)\times \frac{\varepsilon_{a}(\lambda )-1}{\varepsilon_{a}(\lambda )+2}$$
where m indicates that the crystallization fraction of the GST film ranging from 0% to 100%, ε_c_ (λ) and ε_a_ (λ) are the permittivities of GST in the crystalline and amorphous phase, respectively.

Figure 1d shows the dependence of emissivity of the proposed thermal emitter on the crystalline fraction of GST under normal incidence. For the aGST MIM thermal emitter, four resonant emission peaks with average emissivity of 96% can be clearly seen in the simulated spectral emissivity in wavelength range of both 3–5 μm and 8–14 μm atmospheric windows, which are regarded as detected wavelength of IR detectors. These high emissivities do not satisfy the condition of thermal camouflage that requires radiant energy as little as possible in the middle-IR range (3–5 μm) and longer-IR range (8–14 μm). For the cGST MIM thermal emitter, only one broad resonant emission peak with emissivity of near unit at 6.3 μm can be seen, and meanwhile, the emissivity in the range of both 3–5 μm and 8–14 μm is relatively low, meeting the requirements of thermal camouflage. In addition, the near unit emissivity at 6.3 μm with FMHW of 2.7 μm can lead to the enhancement of radiation heat dissipation in the wavelength range 5–8 μm, thereby decreasing the temperature of objects and maintaining the thermal stability of system.

To understand the physical mechanism of the proposed thermal emitter, it is beneficial to study the dependence of absorption on the geometry parameters. In general, the magnetic resonance is stimulated within the dielectric layer by looping displacement current formed antiparallel displacement current excited in the upper and lower metallic film, and it can be attenuated easily with an increasing dielectric layer thickness. Here, the wavelength of magnetic resonance shifts toward longer wavelength with increasing the diameter of gold disk and the permittivity of GST with fixing other geometrical parameters. On the other hand, the wavelength of anti-reflection resonance (it also can be called destructive interference) is directly proportional to the refractive index and thickness of dielectric layer. Consequently, the wavelength of anti-reflection resonance has a red-shift with increasing the refractive index and thickness of GST layer with fixing other geometrical parameters. In conclusion, all these facts are necessary for designing thermal emitter achieving tunable thermal camouflage and radiation heat dissipation. In addition, the spectral emissivity of the MIM thermal emitter based on GST in intermediate phase, has also been investigated theoretically by varying the crystallization fractions of GST from 0% to 100%, showing a tunable and continuous manipulations of the thermal radiation. Therefore, the proposed MIM thermal emitter possessing continuously tunable emissivity shows the ability of adaptive thermal camouflage when the external environment changes. In the following, we will demonstrate that the MIM thermal emitter based on cGST can realize the performance of thermal camouflage in both 3–5 μm and 8–14 μm and maintain thermal stability owing to high emissivity in 5–8 μm.

## 3. Physical Mechanism

To unveil the underlying physical mechanism of different emission peaks, the enhancements of magnetic field component |Hy| in x-z plane have been investigated and presented in Figure 2. It can be observed that the emissivity of MIM thermal emitter based on aGST has four main resonant modes at wavelengths of 3.8 μm, 4.1 μm, 4.6 μm and 12.5 μm, respectively.

The magnetic field at a wavelength of 3.8 μm is confined to the vicinity of interface between the bottom gold layer and the GST layer, which can be generally regarded as anti-reflection resonance [27] (it also can be called destructive interference). The magnetic field of emission peak at 4.1 μm is not only located in the vicinity of interface between the bottom gold layer and the GST layer, but also in the GST layer between bottom layer and the Au top nanodisk. It indicates that this emission peak corresponds to a hybrid-resonance mode, which involves the interaction the third-order magnetic resonance with the anti-reflection resonance. The magnetic fields of emission peaks at both 4.6 μm and 12.5 μm are limited in the GST layer between bottom Au layer and the top Au nanodisk, indicating that a typical third-order magnetic resonance and a fundamental magnetic resonance are generated, respectively. However, the spectral emissivity of the cGST-Emitter possesses merely one emission peak in the interested wavelength range of 5–8 μm. The magnetic field at wavelength of 6.3 μm is localized in the GST layer between the bottom gold layers and the top Au nanodisk, indicating that the hybrid mode including the anti-reflection resonance and third-order magnetic resonance.

These results are in good agreement with Figure 1d, and all the four resonant wavelengths slowly increase as crystallization fraction grows from 0% to 100%. When the GST film is fully crystallized, three resonant peaks in middle-IR range of 3–5 μm for aGST MIM emitter form a single resonant peak with FWHM of 2.7 μm and near unity emissivity at 6.3 μm. In addition, the resonant peak in longer-IR range of 8–14 μm for aGST MIM emitter gradually red-shifts and disappears as crystallization fraction grows from 0% to 100%.

## 4. Results and Discussions

As shown in Figure 1d, our proposed MIM thermal emitter incorporating cGST presents low emissivity in the middle-IR range of 3–5 μm and the longer-IR range of 8–14 μm and high emissivity with a broad FWHM in the wavelength range of 5–8 μm. To evaluate the thermal camouflage performance of the proposed MIM thermal emitter, the IR signal reduction rate $\beta \left(T\right)$ for various surface temperature can be calculated by [29]:(2)$$\beta \left(T\right)=\left(1-\frac{\gamma \left(T\right)}{\delta \left(T\right)}\right)\times 100\%$$
where the $\delta \left(T\right)$ and $\gamma \left(T\right)$ are the radiant intensity of the blackbody and designed MIM thermal emitter at temperature T, respectively. Herein, we only consider the radiant intensity of emitter itself and ignore the ambient radiant intensity and the solar radiant energy reflected by emitter. Spectral radiation intensity of blackbody and emitter can be calculated by [45]:(3)$$E_{b}\left(\lambda ,T\right)=\frac{2\pi hc_{0}^{2}\lambda^{-5}}{\exp\left[hc_{0}/\left(k\lambda T\right)\right]-1}$$
and
(4)$$E_{e}\left(\lambda ,T\right)=\varepsilon \left(\lambda \right)\times E_{b}\left(\lambda ,T\right)$$
respectively, where h is the Planck constant, c_0_ is the speed of light in a vacuum and k is Boltzmann constant. Moreover, $\varepsilon \left(\lambda \right)$ is the spectral emissivity of the MIM thermal emitter. According to Kirchhoff’s law, the emissivity equals to absorptivity under the condition of thermal equilibrium. To separately evaluate $\delta \left(T\right)$ and $\gamma \left(T\right)$, we discretize the equations as a function of wavelength as follows [46]:(5)$$\delta \left(T\right)=\int_{\lambda_{1}}^{\lambda_{2}}E_{b}\left(\lambda ,T\right)d\lambda ≅\sum E_{b}\left(\lambda ,T\right)\Delta \lambda$$
and
(6)$$\gamma \left(T\right)=\int_{\lambda_{1}}^{\lambda_{2}}\varepsilon \left(\lambda \right)E_{b}\left(\lambda ,T\right)d\lambda ≅\sum \varepsilon \left(\lambda \right)E_{b}\left(\lambda ,T\right)\Delta \lambda$$
where λ_1_, λ_2_ are the lower and upper limit of the band, respectively. Using above equations, we could evaluate the radiant intensity in the bands of 3–5 μm, 5–8 μm and 8–14 μm.

In the following, thermal camouflage performance of the MIM thermal emitter based on cGST will be investigated, with the blackbody background temperature varying from 300 K to 800 K. Considering the radiant intensity from the normal direction, Figure 3a gives the spectral radiation intensity of blackbody with and without the emitter, respectively, at temperature of 500 K. Obviously, the cGST-Emitter can effectively tailor blackbody radiation. Figure 3b shows the dependence of signal reduction rate on temperature in the wavelength range of 3–5 μm and 8–14 μm. For the middle-IR ranges of 3–5 μm, signal reduction rates increase slightly and keep above 80% for the temperature from 450 K to 800 K. For the longer-IR range of 8–14 μm, the IR signal reduction rate changes slightly and maintain above 80% for the temperature from 300 K to 800 K.

However, low emissivity in the wavelength ranges of 3–5 μm and 8–14 μm will lead to a decrease in radiation heat dissipation and influence its own temperature. To compensate the decrease of radiation heat dissipation, high emissivity with FWHM of 2.7 μm has been achieved in the 5–8 μm ranges (Figure 3a), and the corresponding radiant intensities for various surface temperatures are depicted in Figure 3b. The radiant intensity in wavelength range of 5–8 μm increases with the emitter temperature going up and this tendency obeys Stefan–Boltzmann law: $E=\varepsilon \sigma T^{4}$ (where ε, σ and T are emissivity, Boltzmann constant and surface temperature, respectively). As a result, the temperature of emitters gradually reduces to equilibrium point and the thermal stability of the emitter can be achieved.

The dependence of the thermal camouflage performance on the different crystallization fraction is further studied at temperature 500 K. Multiple spectral radiation peaks of MIM thermal emitter gradually shift toward middle wavelength IR ranges and finally form single spectral radiation peak in wavelength ranges 5–8 μm when the crystallization fraction of GST from 0% to 100%, as shown in Figure 4a. However, in the wavelength ranges of 3–5 μm, the IR signal reduction rate gradually decreases from 40% to 31% as the crystallization fraction increases from 0 to 0.2 and then slowly increases to 80% when the crystallization fraction increases to 1, as shown in Figure 4b. It is because the radiant intensity in the wavelength of 3–5 μm gradually increases when the crystallization fraction changes from 0 to 0.2. The IR signal reduction rate changes little in wavelength ranges of 8–14 μm and it still keeps above 80%. The radiation intensity in wavelength ranges of 5–8 μm increases when the GST changes from amorphous state to crystalline state, as shown in Figure 4b. In simple words, the MIM thermal emitter based on phase-change material GST with full crystallization (cGST) can simultaneously achieve thermal camouflage and stability.

The dependence of the thermal camouflage performance on the incident angles is also studied. When the incident angle increases from 0° to 25°, the spectral emissivity changes little in the wavelength range of 5–8 μm, but has a bit of enhancement in the wavelength range of 3–5 μm and 8–14 μm, as shown in Figure 5a. Even though the spectral emissivity varies for different incident angles, the requirements of thermal camouflage and stability can be achieved. Figure 5b presents that the IR signal reduction rate decreases with the increased incident angles, but average IR signal reduction rate still keeps above 75% in the wavelength ranges of 3–5 μm and 8–14 μm. Figure 5b shows the radiant intensity in the wavelength of 5–8 μm, which slightly increases when the incident angle increases from 0 to 25°. These results demonstrate that the performance of thermal camouflage and stability is robust to incident angle.

Figure 6a presents that the spectral emissivity almost does not change for different polarization angles, because our designed tunable plasmonic matematerial thermal emitter has fully symmetrical structure. The IR signal reduction rate changes a little with the increasing polarization angles in the wavelength range of 3–5 μm, as shown in Figure 6b. Both IR signal reduction rate in 8–14 μm and radiant intensity in 5–8 μm almost does not change as shown in Figure 6b, respectively. Therefore, the performance of thermal camouflage and stability is also not sensitive for different polarization angles.

As mentioned above, our proposed MIM thermal emitter based on GST is investigated in the blackbody background with temperature equaling to emitter temperature to evaluate the performance of the thermal camouflage device. However, most nonmetallic or metallic environments, such as vegetation (emissivity ε = 0.97), desert (emissivity ε = 0.9), iron (emissivity ε = 0.21), and aluminum (emissivity ε = 0.09) have different emissivities from a blackbody. Furthermore, the temperature of the ambient background is constantly changing, while the temperature of the object remains constant under most circumstances. Herein, our proposed MIM thermal emitter based-on the phase-change material of GST is theoretically demonstrated to continuously hide objects at varying background temperatures and different ambient backgrounds.

The radiant intensities of different background, aGST-Emitter, and cGST-Emitter are shown in Figure 7a,b. The radiant intensity is obtained by integrating the theoretical calculation spectral radiant intensity from 3 μm to 5 μm, which is the working wavelength of IR detector. It can be observed that the tuning range of radiant intensity Δ*P* (*P_max_* − *P_min_*) is the difference in the radiant intensity between the aGST-Emitter and cGST-Emitter. When the radiant intensity of the background is in the range from *P_min_* to *P_max_*, continuous thermal camouflage can be achieved by controlling the GST crystallization fraction, to meet the requirement of radiant intensity of the MIM thermal emitter equal to radiant intensity of the background. When the radiant intensity of the background is *P_max_*, the upper limit of the background temperature is *T_max_*. Similarly, the lower limit of the background temperature corresponding to *P_min_* is *T_min_*. Therefore, the temperature range of background camouflage Δ*T* is the difference between *T_max_* and *T_min_*. In addition, the object can be camouflaged over a wide range of background temperatures by only increasing the object temperature.

As shown in Figure 7a, if the emitter temperature is 60 °C, thermal camouflage can be achieved when the vegetation and desert background temperature ranges from 20 °C to 45 °C. As shown in Figure 7b, the radiant intensity of the cGST-Emitter and the metallic background made of iron are nearly equal in temperature ranges from −20 °C to 40 °C. Therefore, the cGST-Emitter has the ability to continuously camouflage objects at varying background temperatures when background is iron. The camouflage temperature ranges of aluminum background are from 60 °C to 100 °C, when the thermal emitter temperature is 30 °C. It benefits camouflaging the object in extreme ambient backgrounds with a large temperature range.

Simulated IR images demonstrate that our proposed tunable thermal emitters can continuously achieve thermal camouflage at different background temperatures and ambient background. When the vegetation background temperature is 20 °C, the radiation temperature of both cGST-Emitter and vegetation background is approximately 19 °C, and therefore the cGST-Emitter is difficult to distinguish from background in the IR image, as shown in Figure 7e. In Figure 7h, when the temperature of vegetation background increases from 20 °C to 45 °C, the cGST-Emitter of low radiation temperature can be easily distinguished from the hot background, since the radiation temperature of the cGST-Emitter remains at approximately 19 °C. As shown in Figure 7l, the radiation temperature of the aGST-Emitter matches with that of desert background (45 °C), achieving thermal camouflage. When the background temperature is 20 °C, the hot aGST-Emitter can be easily observed from the cold background, since the radiation temperature of the aGST-Emitter remains at approximately 45 °C in Figure 7i. Similarly, when the background is aluminum and iron, Figure 7q,r,w,s are perfect thermal camouflage cases at different background temperatures.

## 5. Conclusions

In summary, a tunable thermal emitter, consisting of MIM plasmonic metamaterial based on GST, is proposed to realize tunable manipulations of thermal radiation and thermal camouflage, owing to low emissivity in both atmospheric windows. Meanwhile, the tunable thermal emitter possesses near unity emissivity in 6.3 μm to compensate for decreasing radiation heat dissipation, which can maintain the thermal stability of the system. The underlying mechanism relies on fundamental magnetic resonance and coupling between the high-order magnetic resonance and anti-reflection resonance. When the environmental background is blackbody, the tunable emitter maintains middle-IR signal reduction rates greater than 80% from 450 K to 800 K, and longer-IR signal reduction rates greater than 80% from room temperature to 800 K. In addition, our proposed thermal camouflage GST device theoretically demonstrates the ability to continuously modulating camouflaging objects at varying background temperatures and differently environmental background. This ultrathin plasmonic metamaterial thermal emitter based on phase-change material of GST provides a scheme of worthy reference for thermal emission control, radiative cooling, infrared detection, thermophotovoltaics and tunable thermal camouflage.

## Figures and Tables

**Figure 1 nanomaterials-11-00260-f001:**
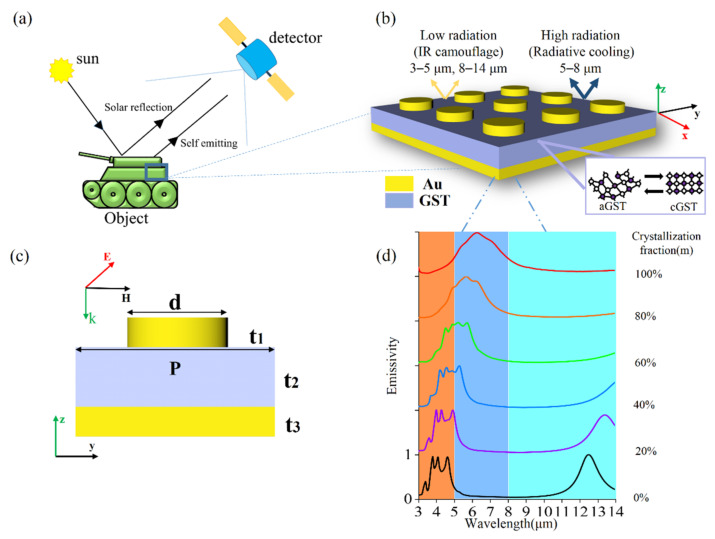
(**a**) Schematic of the IR detection process and a typical application for a tank requiring thermal camouflage; (**b**) Schematic and (**c**) sectional view of the proposed ultrathin MIM plasmonic thermal emitter based-on phase-change material GST. The MIM thermal emitter consists of the top gold nanodisks, the intermediate GST layer and the bottom gold layer; (**d**) The simulated spectral emissivity of the plasmonic thermal emitter gradually changes as crystallization fraction of GST increases from 0% to 100%.

**Figure 2 nanomaterials-11-00260-f002:**
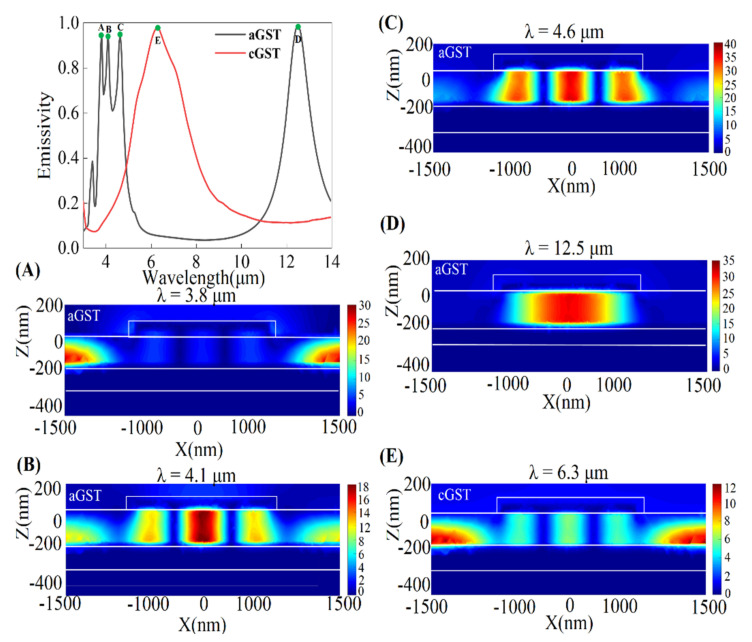
Simulated spectral emissivity of the tunable MIM thermal emitter is obtained when crystallization fraction of GST is 0% and 100%, respectively. The black and red lines represent the aGST-Emitter and cGST-Emitter, respectively. (**A**–**D**) represent the enhancement of magnetic field intensities component |Hy| of the peak length of different resonant modes of the aGST-Emitter. The color-map (**E**) represents the enhancement of magnetic field intensities component |Hy| of single peak length of the cGST-Emitter.

**Figure 3 nanomaterials-11-00260-f003:**
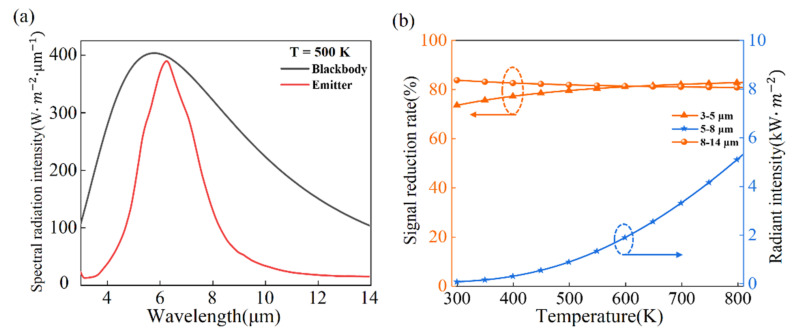
(**a**) The spectral radiation intensity of blackbody and emitter at 500 K, the black and red lines represent the blackbody and the cGST-Emitter, respectively; (**b**) Signal reduction rates of our designed cGST-Emitter. The carmine lines indicate the middle-IR ranges of 3–5 μm and the longer-IR ranges 8–14 μm, respectively. The radiant intensity of cGST thermal emitter in the wavelength range of 5 μm–8 μm is indicated by the blue line for various surface temperatures.

**Figure 4 nanomaterials-11-00260-f004:**
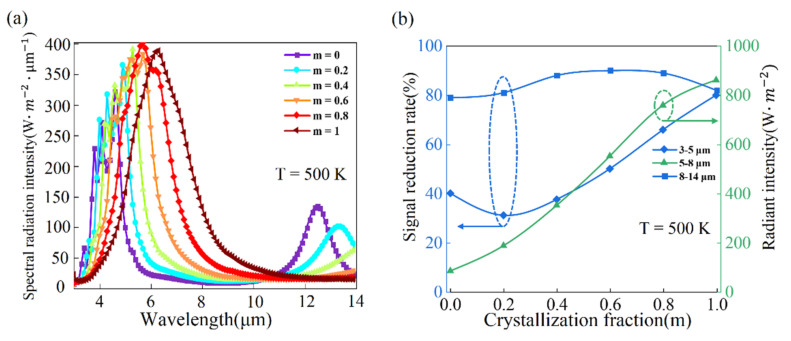
Tunability of the MIM thermal emitter as changing crystallization fraction of GST. (**a**) Spectral radiation intensity of the thermal emitter at corresponding various crystallization fraction in the normal direction at 500 K; (**b**) The IR signal reduction rates in wavelength ranges 3–5 μm and 8–14 μm and the radiant intensity in wavelength ranges 5–8 μm at 500 K for different crystallization fraction.

**Figure 5 nanomaterials-11-00260-f005:**
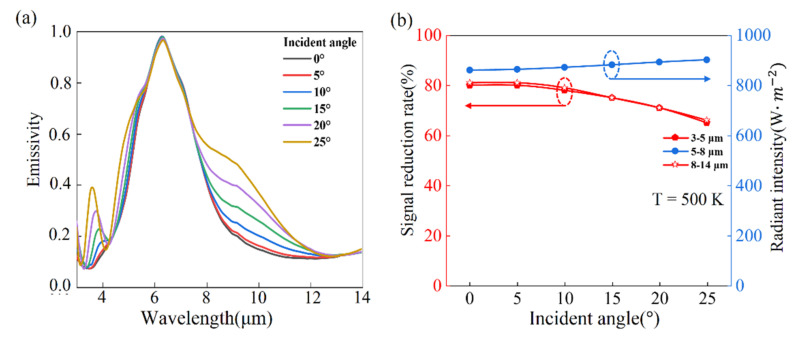
(**a**) Spectral emissivity of cGST thermal emitter in the incident angle changing from 0° to 25°; (**b**) is IR signal reduction rates of our designed cGST thermal emitter in wavelength ranges of 3–5 μm and 8–14 μm at 500 K for various incident angles, respectively. The radiant intensity in wavelength ranges 5–8 μm at 500 K for different incident angles.

**Figure 6 nanomaterials-11-00260-f006:**
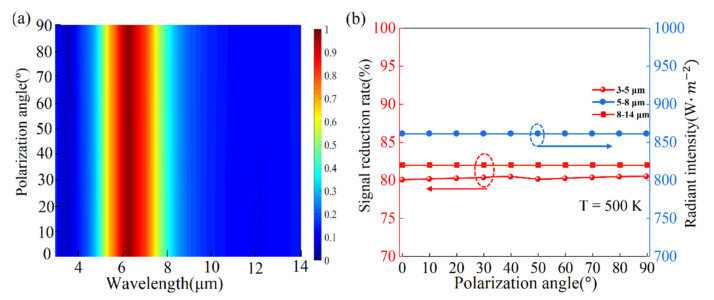
(**a**) Spectral emissivity of cGST thermal emitter for different polarization angles, which change from 0° to 90°; (**b**) is IR signal reduction rates of our designed cGST thermal emitter in wavelength ranges 3–5 μm and 8–14 μm at 500 K for various polarization angles, respectively. The radiant intensity (W/m^2^) in the wavelength 5–8 μm for various polarization angles.

**Figure 7 nanomaterials-11-00260-f007:**
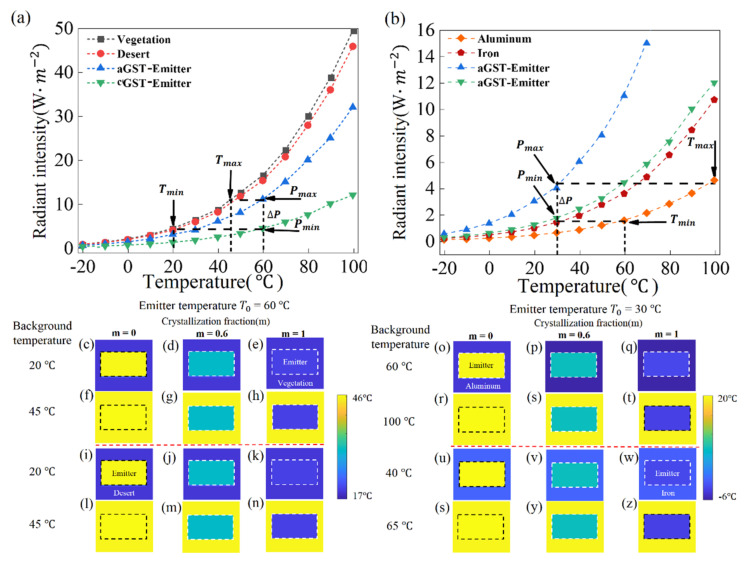
(**a**) The radiant intensity of the nonmetallic background environment including vegetation and desert, aGST-Emitter device, cGST-Emitter device at different temperatures in wavelength range of 3–5 μm; (**b**) The radiant intensity of the metallic background environment including Al and Fe, aGST-Emitter device, cGST-Emitter device at different temperatures in wavelength range of 3–5 μm. (**c**–**z**) IR images are simulated at different background temperatures and environmental background; (**c**–**n**) the fixed emitter temperature is 60 °C at different crystallization fraction of GST (m = 0, 0.6, 1) and background is vegetation in above of red dotted line. On the contrary, background is desert in the bottom of red dotted line; (**o**–**z**) the fixed emitter temperature is 30 °C at different crystallization fraction of GST (m = 0, 0.6, 1) and background is aluminum in above of red dotted line. In contrast, background is iron in the bottom of red dotted line.

## Data Availability

Data available in a publicly accessible repository.

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
