# Peer review of "Tunable Thermal Camouflage Based on GST Plasmonic Metamaterial"

_nanomaterials, 2021, doi:10.3390/nano11020260_

Round 1

Reviewer 1 Report

The submitted manuscript reports on the solely theoretical investigation of tuneable thermal emitter designed to control in a dynamical way its thermal radiation emission for camouflage applications in the 3-5 um and 8-14 um IR domains. The MIM-based photonic gold metastructure is integrating an active layer based on the GST phase change material allowing, through proper designed dimensions and specific amorphous-to-crystalline ratios in the GST layer to generate particular magnetic resonances manifesting in distinct absorption/ emissivity peaks in the infrared regions of interest.Overall, the manuscript is well written, correctly citing previous research in the domain and giving sufficient consistency to the employed methods and theoretical results. Still, I some strong remarks concerning the manuscript, at methodological and fundamental levels:

1.      The methodology and main design ideas and purposes do not bring much novelty compared with previous reports on similar concepts (employing similar phase change materials) like the one reported in Ref. 11 in the manuscript (Pan, M.; Huang, Y.; Li, Q.; Luo, H.; Zhu, H.; Kaur, S; Qiu, M. Multi-band middle-infrared-compatible camouflage with thermal management via simple photonic structures. Nano Energy. 2020, 69, 104449.). The authors in Refs. 11, using a similar theoretical approach and design, have also experimentally demonstrated the feasibility of their proposed devices.

2.      I have a major concern when it comes to the use of GST in such ”dynamical”, passive structures. Although from a theoretical point of view the results are correct, in practical devices, the behaviour of phase change materials is of non-volatile type. Following the heating (note also that the GST will reach its crystalline-state m=1 at temperatures higher than 150°C), the material will therefore retain is state (crystalline or intermediary amorphous-crystalline) when cooling back to the initial temperature. With no additional stimuli (electrical/ optical) employed to bring the GST back to its former phase, the system integrating it will behave as a “one-shot” device, with limited use. From this point of view, the employment of phase change/ phase transition materials with volatile-type behaviour (like e.g., VO2 compounds) may bring a real advantage over the use of GST-type in such passive devices. 3.      From the technical point of view, I do not understand why the signal reduction rate curve in Fig. 6b, for the 8-14 um domain reaches 95% for all the polarisation angles, completely different from the results presented previously on Figs. 3-5. 

The main considerations given above would make difficult, in my opinion, the acceptance of the manuscript for publication in the present form.

Author Response

The attached file.

Reviewer 2 Report

In this work, "Tunable thermal camouflage based on GST plasmonic metamaterial", the authors investigated a tunable thermal emitter including MIM plasmonic metamaterial based on GST. The dependency of thermal camouflage on crystallization fraction of GST, incident light angle, and polarization angle of light have been discussed in detail. Based on the obtained results, they claimed that the thermal emitter can continuously keep thermal camouflage for different background temperatures and environmental conditions between 3-5 um. Overall, this manuscript has a strong potential for a second review after applying the issues and addressing the shortcomings listed below:

1-The authors should polish/revise some grammatical mistakes and typos along the manuscript. To this end, I invite the authors to read their manuscript carefully and make the required changes where necessary.

2-Any particular reason to use Au for the proposed platform, which is not cost-efficient?

3-In the Introduction section, while discussing phase change materials (PCMs) (especially GST) and their possible application areas, the following works should also be considered and cited, to give a more general view to the possible readers of the work: [(i) Extracting the temperature distribution on a phase-change memory cell during crystallization, Journal of Applied Physics 120, 164504 (2016); (ii) The role of Ge2Sb2Te5 in enhancing the performance of functional plasmonic devices, Materials Today Physics 12, 100178 (2020)].

4-In Figure 2, rather than plotting normalized H-field distributions, plot H-field enhancement with the actual maximum values to be able make a comparison at different wavelengths. 

5-In Figure 1b (inset), specify each crystal order as ‘aGST’ and ‘cGST’. 

6-Update the size of Figures 3b, 4b, and 5b. In their current form, they seem like they are squeezed along x-axis.

7-What is the effect of possible gold diffusion in GST to the overall performance of the proposed system? It should be discussed more in detail. 

Author Response

The attached file.

Reviewer 3 Report

In this manuscript a tunable thermal emitter, consisting of MIM metamaterial based on phase-change material GST, is theoretically investigated.

The manuscript presents several new results as those in Figures 4 and 7 and, therefore, could be of interest to readers.

  • The simulations well demonstrate the potential for tunability; nevertheless, the “dynamic” characteristics is not demonstrated, since this is, as the authors state, a theoretical work and the behavior on cyclablity of the phase change is not demonstrate and/or discussed. Therefore, the authors are suggested not to give emphasis to the “dynamics” or “dynamically control of…” and to change those expressions.
  • In section 2. Structure design, the authors are suggested to give a rationale for the chosen geometry of the MIM structure discussed, and which main geometrical parameters affect the resonance in Fig. 2.

Minor comment:

-modify “phase changing” with “phase-change”.

Author Response

The responses lies in the Attached file!

Round 2

Reviewer 1 Report

  1. In spite of author’s explanations, I do not feel like the proposed design bring major novelty (both finality and methodology) and is remaining just a theoretical variant to the one presented in Ref. 11.
  2. The explanations concerning the possibility of reversible transition for GST is at least naïve (“Here, we’d like to state that with the development of science and technology, the stimulation for the GST’s reversible switching is easier and easier.”)   I would strongly expect a proposition for a realistic implementation of an electrical or optical thermal stimulus within the proposed structure instead of general assertions on the facility of reversible switching of GST.

Author Response

(The authors gave the same response as above.)

Reviewer 2 Report

In its current form, the revised manuscript is suitable for publication.

Author Response

(The authors gave the same response as above.)
